# Breaking the $n^{1.5}$ Additive Error Barrier for Private and Efficient Graph Sparsification via Private Expander Decomposition

Anders Aamand [*1]   Justin Chen [*2]   Mina Dalirrooyfard [*3]   Slobodan Mitrović [*4]   Yuriy Nevmyvaka [5]
Sandeep Silwal [*6]   Yinzhan Xu [*7]

## Abstract

We study differentially private algorithms for graph cut sparsification, a fundamental problem in algorithms, privacy, and machine learning. While significant progress has been made, the best-known private and efficient cut sparsifiers on $n$-node graphs approximate each cut within $\widetilde{O}(n^{1.5})$ additive error and $1 + \gamma$ multiplicative error for any $\gamma > 0$ [Gupta, Roth, Ullman TCC'12]. In contrast, *inefficient* algorithms, i.e., those requiring exponential time, can achieve an $\widetilde{O}(n)$ additive error and $1 + \gamma$ multiplicative error [Eliáš, Kapralov, Kulkarni, Lee SODA'20]. In this work, we break the $n^{1.5}$ additive error barrier for private and efficient cut sparsification. We present an $(\varepsilon, \delta)$-DP polynomial time algorithm that, given a non-negative weighted graph, outputs a private synthetic graph approximating all cuts with multiplicative error $1 + \gamma$ and additive error $n^{1.25+o(1)}$ (ignoring dependencies on $\varepsilon, \delta, \gamma$).

At the heart of our approach lies a private algorithm for expander decomposition, a popular and powerful technique in (non-private) graph algorithms.

---

[*]Equal contribution   [1]University of Copenhagen, BARC, Denmark. aa@di.ku.dk.   [2]Massachusetts Institute of Technology, Cambridge, MA, USA. justc@mit.edu.   [3]Machine Learning Research, Morgan Stanley, Canada. minad@mit.edu.   [4]University of California, Davis, USA. smitrovic@ucdavis.edu.   [5]Machine Learning Research, Morgan Stanley, USA. yuriy.nevmyvaka@morganstanley.com.   [6]UW-Madison, USA. silwal@cs.wisc.edu.   [7]University of California, San Diego, USA. Correspondence to: Mina Dalirrooyfard <minad@mit.edu>.

*Proceedings of the 42nd International Conference on Machine Learning*, Vancouver, Canada. PMLR 267, 2025. Copyright 2025 by the author(s).

## 1. Introduction

Many modern applications rely on personal data, with notable examples including social networks and medical data analysis. However, traditional algorithms, developed over decades, are inherently non-private and often sensitive to even small changes in input. This sensitivity can lead to significant privacy breaches (Backstrom et al., 2007; Narayanan & Shmatikov, 2008; Culnane et al., 2019). These privacy concerns have driven extensive research on algorithms that protect users' privacy. Graphs, which are ubiquitous in machine learning and data processing, have received significant attention in privacy-preserving settings. Our work contributes to this growing body of research by designing private algorithms in the context of graph cuts. In social networks, cut queries are crucial in analyzing how tightly a community is connected internally and how it interacts with the outside world. However, accurately releasing connectivity information can compromise user privacy (Hay et al., 2009). Moreover, there are exponentially many cuts, making a direct private release of all the cuts infeasible. Hence, one way to solve this problem is to construct a synthetic graph on the same node set as the original while preserving approximate cut values.

The standard notion of privacy is *differential privacy (DP)* introduced by Dwork, McSherry, Nissim, and Smith in their seminal work (Dwork et al., 2006), which indicates that the output of a private algorithm of two neighboring inputs must be statistically indistinguishable. Formally, an algorithm $A$ is $(\varepsilon, \delta)$-DP if given two *neighboring* inputs $G$ and $G'$ and a subset of outputs $O$, we have $\Pr(A(G) \in O) \le e^{\varepsilon} \Pr(A(G') \in O) + \delta$. When $\delta = 0$, we say that the algorithm preserves *pure*-DP, and otherwise *approximate*-DP. In the context of graphs, the most widely studied notion of neighboring graphs is *edges-neighboring*: two neighboring graphs differ in only one edge, and the graphs' node sets are publicly available. For weighted graphs, two neighboring graphs are those whose total edge weights differ by at most one, and in a single edge[1].

---

[1]Algorithms satisfying this notion are often also private for graphs whose vector of $\binom{n}{2}$ edge weights differ by at most 1 in $\ell_1$ distance.

Given a non-negative weighted, undirected graph $G = (V, E, w)$, we consider the problem of releasing a non-negative weighted, undirected synthetic graph $\widetilde{G}$ on the same node set $V$ which (a) is differentially private and (b) approximately preserves values of all the cuts in $G$. A cut is formed by a subset $S \subset V$ where $S \neq \emptyset$, and the value of the cut $w_G(S)$ is the weight of the edges between $S$ and $V \setminus S$. Prior works (Gupta et al., 2012; Eliáš et al., 2020; Liu et al., 2024) have settled the complexity of this problem when *no multiplicative error is allowed*. In particular, Gupta, Roth, and Ullman showed how to release a synthetic graph in a pure-DP manner and with $O(n^{1.5})$ additive error of each cut value. When the number of edges of the input graph is upper-bounded by $m$, the additive error can be improved to $\widetilde{\Theta}(\sqrt{mn})$ (Eliáš et al., 2020; Liu et al., 2024) for approximate-DP algorithms.[2] Thus, despite a long line of work, the additive error of this problem has been stuck at $O(n^{1.5})$ for dense graphs, the same error that the simple algorithm of just adding Laplace noise to all possible edges achieves (although this would result in a graph with negative edges, whereas the aforementioned works release a non-negative weight graph). The fundamental reason for this is that $\Omega(n^{1.5})$ additive error is necessary even for approximate-DP (Eliáš et al., 2020; Liu et al., 2024).

The synthetic graph that achieves the $O(n^{1.5})$ bound is a dense graph. However, in many applications, storing only a sparse graph is desirable due to communication or memory requirements. Indeed, this is the original motivation for the extensive algorithmic work on graph sparsification (going back to (Karger, 1994) more than three decades ago). This requirement is also crucial in many other sub-fields, such as the semi-streaming graph model (Feigenbaum et al., 2005). For sparse synthetic graphs, it is unavoidable to also have a multiplicative error in addition to the additive error (even non-private graph sparsification has the multiplicative error). However, the $\Omega(n^{1.5})$ lower bound does not apply when a multiplicative error is allowed. In fact, using the exponential mechanism, it is known how to achieve multiplicative error $1 + \gamma$ and additive error $O(n \log n)$ (Eliáš et al., 2020), which is known to be near-optimal (Dalirrooyfard et al., 2023). One caveat, though, is that this algorithm takes *exponential time*, and the best polynomial time algorithm still has $O(n^{1.5})$ additive error. Closing this gap between $O(n^{1.5})$ and $\widetilde{O}(n)$ additive error, while allowing for $1 + \gamma$ multiplicative approximation, is considered "a prominent open problem in the differential privacy literature" (Eliáš et al., 2020).

In this work, we provide the first polynomial time algorithm that beats the $O(n^{1.5})$ additive error barrier, thus making

---

[2] Throughout this paper, $\widetilde{O}, \widetilde{\Theta}, \widetilde{\Omega}$ hide $\mathrm{polylog}(n/\delta)$ factors for approximate-DP problems, and $\mathrm{polylog}(n)$ factors for other problems.

substantial progress toward resolving this open problem.

**Theorem 1.1.** *Let* $\gamma \in (0, 1)$. *There is a polynomial time* $(\varepsilon, \delta)$-*DP algorithm, which on any non-negative weighted $n$-node graph $G = (V, E, w)$ outputs a non-negative weighted sparse graph $H$ with $\widetilde{O}(n/\gamma^2)$ edges such that with high probability:*

$$\forall S \subseteq V, \quad |w_G(S) - w_H(S)|$$
$$\leq \gamma w_G(S) + \frac{n^{1.25+o(1)} \, \mathrm{polylog}(1/\delta)}{\varepsilon \gamma^{0.5}}.$$

We remark that prior work listed in Table 1 (except the first one which adds Laplace noise to every edge) can also output a sparse graph by simply running a non-private sparsifier in post-processing. Then the prior results in Table 1 also incur the same multiplicative approximation as Theorem 1.1 (but retain their additive error). Thus, the best prior work on *sparse* cut-sparsifiers with $\widetilde{O}(n/\gamma^2)$ edges has multiplicative approximation $1 + \gamma$ and additive error $O(n^{1.5}/\varepsilon)$ (Gupta et al., 2012), even allowing approximate DP.

A key technical ingredient for proving Theorem 1.1 is private expander decomposition, which may be of independent interest; see more details in Section 4. The formal proof of Theorem 1.1 can be found in Section 3. We show that Theorem 1.1 has important downstream applications in Section 5 and Appendix B, where we obtain DP algorithms for max-cut, maximum-bisection, max-$k$-cut, and minimum-bisection, with optimal multiplicative and improved additive error in the DP setting. We conclude in Section 6 with several open questions that we find exciting and interesting to pursue.

### 1.1. Related Work

Here, we mention some relevant problems to our work. There has been a lot of work on releasing synthesized graphs in a DP manner with objectives other than cut queries, such as maintaining the degree sequence or subgraph counting (see the survey (Li et al., 2023)). Depending on the objective, different techniques are used. There have been studies on DP algorithms for other cut problems such as min cut (Gupta et al., 2010), min-$st$-cut (Dalirrooyfard et al., 2023), all pairs min-$st$-cut (Aamand et al., 2024) and multiway cut (Dalirrooyfard et al., 2023; Chandra et al., 2024), where the error guarantees for all these problems are optimal. Note that the output size in these problems is polynomial, so there is no need to output a synthetic graph, and the techniques are not directly applicable to our problem.

### 1.2. Technical Contribution

Our main technical contribution is an algorithm which, given an input graph $G = (V, E, w)$, in a differentially private manner outputs a synthetic graph $\widetilde{G}$ such that for any

| Reference | DP | Multiplicative Error | Additive Error | Runtime |
|---|---|---|---|---|
| Laplace Noise* | Pure | 1 | $O(n^{1.5}/\varepsilon)$ | Polynomial |
| (Gupta et al., 2012) | Pure | 1 | $O(n^{1.5}/\varepsilon)$ | Polynomial |
| (Arora & Upadhyay, 2019) | Approx | $1 + \alpha$ | $\widetilde{O}(\sqrt{n}|S|/\varepsilon)$ | Polynomial |
| (Eliáš et al., 2020; Liu et al., 2024) | Approx | 1 | $\widetilde{O}(\sqrt{mn}/\varepsilon)$ | Polynomial |
| (Eliáš et al., 2020) | Pure | $1 + \alpha$ | $O(n \log n/\varepsilon)$ | Exponential |
| **Theorem 1.1 (Our Work)** | Approx | $1 + \alpha$ | $n^{1.25+o(1)}/\varepsilon$ | Polynomial |

*Table 1.* A comparison of approaches for differentially-private cut-approximation that release synthetic graphs. Dependencies on the approximate DP parameter $\delta$ are hidden, but the dependency is only $\mathrm{polylog}(1/\delta)$ in all cases. We also omit the dependency on the multiplicative approximation parameter $\alpha$. (Arora & Upadhyay, 2019) obtain a bound of $\widetilde{O}(\sqrt{n}|S|/\varepsilon)$ to approximate a $(S, V \setminus S)$ cut, which can be $\widetilde{\Omega}(n^{1.5})$ for large $|S|$. We note that $\Omega(n^{1.5})$ additive error is necessary even for approximate-DP (Eliáš et al., 2020; Liu et al., 2024) if no multiplicative error is allowed.
*The first result of naively adding Laplace noise does not produce a non-negative graph, which the other results ensure.

cut $(C, V \setminus C)$ we have $|w_G(C) - w_{\widetilde{G}}(C)| \leq \alpha w_G(C) + n^{1.25+o(1)}$ for any desired small constant $\alpha = \Omega(1)$. To obtain a sparse graph approximating all cuts with a similar approximation guarantee, we then run any graph sparsification algorithm on $\widetilde{G}$, e.g., the classic algorithm by Benczúr and Karger (1996), and by post-processing properties of differential privacy, the resulting graph is still private.

To describe our construction of $\widetilde{G}$, we first outline two other approaches for private graph cut approximation. For the sake of simplicity, throughout this overview, we assume the parameters $\varepsilon, \delta, \alpha$ are all constants.

**Approach 1: Additive error of $\widetilde{O}(\sqrt{nm})$.** For approximating all cuts in a DP-manner with purely additive error guarantees Eliáš, Kapralov, Kulkarni, and Lee (2020) provide an $(\varepsilon, \delta)$-DP algorithm with an additive approximation of $\widetilde{O}(\sqrt{nW})$ for graphs with total edge weights bounded by $W$. Liu, Upadhyay, and Zou (2024) provide an improved algorithm with approximation $\widetilde{O}(\sqrt{nm})$. In the case where the number of edges $m$ is *small*, these algorithms already achieve an improvement over additive error $O(n^{1.5})$.

**Approach 2: Additive error of $\widetilde{O}(|C|\sqrt{n})$.** Another way of privatizing $G$ is to add Laplace noise $\mathrm{Lap}(1/\varepsilon)$ to the weight of each edge of $G$. This algorithm is trivially private, and by standard concentration bounds and union bounding over all cuts of size $s$, one can see that these cut values are preserved within additive error $\widetilde{O}(s\sqrt{n})$. In fact, Upadhyay, Upadhyay, and Arora (2021) provide an approximate DP algorithm with essentially the same approximation guarantee and with the additional property that the synthetic graph they output has *non-negative* edge weights. For cuts $C$ where $s = |C| = o(n)$, these algorithms again improve over the additive error of $O(n^{1.5})$.

**Our approach.** Our approach is illustrated in Figure 1. The main observation behind our algorithm is the following: If for all cuts $(C, V \setminus C)$ the cut weight $w_G(C)$ is $\widetilde{\Omega}(|C|\sqrt{n}/\alpha)$, then the $\widetilde{O}(|C|\sqrt{n})$ additive error achieved by Approach 2 is only an $\alpha$ fraction of $w_G(C)$. Phrasing it differently, the synthetic graph output by Approach 2 already achieves a $(1 + \alpha)$-multiplicative approximation for all the cuts.

Unfortunately, not all graphs satisfy the property described above, i.e., $w_G(C) \in \widetilde{\Omega}(|C|\sqrt{n}/\alpha)$. Nevertheless, that observation raises the following question: *Can we process an input graph $G$ so that the resulting graph, or conveniently chosen subgraphs of $G$, exhibit that desired property?* Our approach answers this question affirmatively and shows how to leverage it to obtain the advertised upper bound on the additive error. We now provide more details.

**Graph expanders.** Graphs satisfying the above property coincide with the notion of expander graphs.[3] To elaborate, for a cut $(C, V \setminus C)$ of the graph, we define its sparsity as

$$\phi(C) = \frac{w(C)}{\min\{|C|, |V \setminus C|\}}.$$

The sparsity of the graph is then defined as $\phi(G) = \min_{\emptyset \subsetneq C \subsetneq V} \phi(C)$. If $\phi(G) > \psi$, we call $G$ a $\psi$-expander. The property considered in the previous paragraph can be phrased as $\phi(G) \geq \widetilde{\Theta}(\sqrt{n}/\alpha)$.

Expander decomposition is a popular and powerful algorithmic tool that has been studied by numerous groups, e.g., (Nanongkai & Saranurak, 2017; Wulff-Nilsen, 2017; Saranurak & Wang, 2019; Chuzhoy et al., 2020; Li & Saranurak, 2021). For any parameter $\psi > 0$, it is known how to decompose the vertices $V$ of an arbitrary weighted graph into

---

[3]There are several notions of expanders, including edge, vertex, and spectral expanders. We use a certain version of edge expanders.

$V_1, \ldots, V_k$, so that the induced subgraph $G[V_i]$ for every $V_i$ is a $\psi$-expander, and the sum of weights of the edges that are not within any $V_i$ is $\widetilde{O}(n\psi)$. Suppose we can privately obtain such a decomposition for $\psi = \widetilde{\Theta}(\sqrt{n}/\alpha)$. Then, the sum of weights of the inter-component edges – that we refer to by $E_{\mathrm{sparse}}$ – would be $\widetilde{O}(n^{1.5}/\alpha)$. We apply Approach 1 to $E_{\mathrm{sparse}}$ to obtain a synthetic graph that preserves each cut with an additive error of $\widetilde{O}(\sqrt{n^{1.5}/\alpha \cdot n}) = \widetilde{O}(n^{1.25}/\sqrt{\alpha})$. For the edges within each $G[V_i]$, we apply Approach 2 to obtain a private graph sparsifier that preserves each cut in $G[V_i]$ with $(1 + \alpha)$-multiplicative approximation.

**Private expander decomposition.** Unfortunately, to the best of our knowledge, private expander decomposition has not been studied in the literature.

One natural approach is to execute a non-private expander decomposition algorithm on a private synthetic graph $\widetilde{G}$, e.g., $\widetilde{G}$ obtained by Approach 2. However, this straightforward approach has a major issue: Let $w_G(S, T)$ be the sum of the weights of edges between $S$ and $T$ for any $S, T \subset V(G), S \cap T = \emptyset$. If $\widetilde{G}[V_i]$ is a $\widetilde{\Theta}(\sqrt{n})$-expander, it does not immediately imply that $G[V_i]$ is a $\widetilde{\Theta}(\sqrt{n})$-expander. In order to achieve so, one would need to lower-bound $\frac{1}{|C|} \left| w_G(C, V_i \setminus C) - w_{\widetilde{G}}(C, V_i \setminus C) \right|$ by $\widetilde{\Theta}(\sqrt{n})$, which the algorithm in Approach 2 does not achieve; it achieves this bound for $V = V_i$.[4]

Hence, instead of applying non-private expander decomposition algorithms in a black-box way, we privatize an existing expander decomposition algorithm in a white-box manner. On a high level, our approach follows the work by (Nanongkai & Saranurak, 2017). To support weighted graphs, we replace one of the key subroutines in (Nanongkai & Saranurak, 2017) with a result from (Li & Saranurak, 2021). As the outcome, we obtain a private expander decomposition algorithm that, for $\psi \geq n^{0.5+o(1)}$, decomposes a graph into $\psi$-expanders with the total weights of inter-component edges upper-bounded by $\psi \cdot n^{1+o(1)}$. Our private expander decomposition might be of independent interest.

## 2. Preliminaries

**Definition 2.1** (Cut crossing edges). Let $G = (V, E, w)$ be a weighted graph and $S \subset V$ a subset of vertices. We use $w_G(S)$ to denote the total weights of edges with one endpoint in $S$ and the other endpoint in $V \setminus S$. That is,

$$w_G(S) = \sum_{\{u,v\} \in E: u \in S, v \in V \setminus S} w(u, v).$$

---

[4]Such bounds are achievable if we allow negative weights in $\widetilde{G}$ (Upadhyay et al., 2021), but then we will not be able to run non-private expander decomposition algorithms on $\widetilde{G}$ because they only work for graphs with non-negative edge weights.

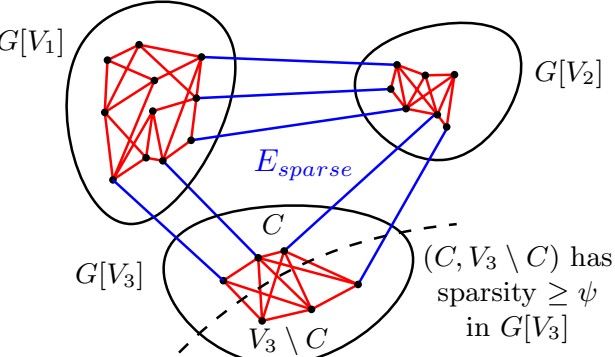

*Figure 1.* For $\psi = n^{0.5+o(1)}$, our algorithm privately obtains a partition of the vertices of $G$ into sets $V_1, \ldots, V_k$ such that (1) each induced subgraph $G[V_i]$ (red) has sparsity at least $\psi$ and (2) the total weight of edges between components (blue) is $\widetilde{} n^{1+o(1)}$. We run the $\widetilde{O}(\sqrt{nm})$ additive error algorithm by (Eliáš et al., 2020) on $E_{\mathrm{sparse}}$ formed by the blue edges, and the algorithm by (Upadhyay et al., 2021) on each $G[V_i]$. The high density of $G[V_i]$ ensures that the multiplicative approximation can subsume the error of the latter algorithm.

When it is clear from the context, we omit the subscript $G$.

**Theorem 2.2** ((Upadhyay et al., 2021)). *Given a non-negative weighted $n$-node graph $G = (V, E)$, there is a polynomial time $(\varepsilon, \delta)$-DP algorithm which outputs a non-negative weighted graph $\widetilde{G}$ on the same vertex set $V$, such that with probability $1 - \lambda$, for any $S \subseteq V$,*

$$\left| w_G(S) - w_{\widetilde{G}}(S) \right| \leq O \left( \frac{|S| \log(\frac{1}{\lambda}) \sqrt{n \log n \log(\frac{1}{\delta})}}{\varepsilon} \right).$$

To be more precise, we use the synthetic graph $\bar{G}$ described in their Algorithm 8, and the guarantee on the cut values is implied by their Equation (9). In their paper, they only formally state the results for the case where $\lambda$ is a constant, but they can be easily extended to arbitrarily small $\lambda$ by incurring a cost of $O(\log(1/\lambda))$ (see their discussion after Theorem 15).

**Theorem 2.3** ((Eliáš et al., 2020)). *Given a non-negative weighted $n$-node graph $G = (V, E, w)$, there is a polynomial time $(\varepsilon, \delta)$-DP algorithm for $0 < \varepsilon < 1/2$ and $0 < \delta < 1/2$, which outputs a non-negative weighted graph $\widetilde{G}$ on the same vertex set $V$, such that with high probability, for every $S \subseteq V$,*

$$\left| w_G(S) - w_{\widetilde{G}}(S) \right|$$
$$\leq O \left( \sqrt{\frac{w(E) n \log n}{\varepsilon}} \log^2 \left( \frac{n \log n}{\delta} \right) \right),$$

*where $w(E)$ denotes the sum of all edge weights.*

The original statement in (Eliáš et al., 2020) does not provide a high probability guarantee, so we provide a proof in Appendix A for the above version.

**Differential Privacy Tools.** We now list basic definitions and tools we use in the context of differential privacy.

**Definition 2.4** (Edge-Neighboring Graphs). Graphs $G = (V, E, w)$ and $G' = (V, E', w')$ are called *edge-neighboring* if there is $(u, v) \in V^2$ such that $|w(u, v) - w'(u, v)| \leq 1$ and for all $\{u', v'\} \neq \{u, v\}$, $u', v' \in V^2$, we have $w(u', v') = w'(u', v')$. Non-edges are considered to have zero weight.

**Definition 2.5** (Differential Privacy (Dwork, 2006)). A randomized algorithm $\mathcal{A}$ is $(\varepsilon, \delta)$-DP if for any neighboring graphs $G$ and $G'$ and any set of possible outcomes $O$ of $\mathcal{A}$ it holds $\Pr(\mathcal{A}(G) \in O) \leq e^\varepsilon \Pr(\mathcal{A}(G') \in O) + \delta$.

When $\delta = 0$, algorithm $\mathcal{A}$ is called *pure DP*, or only $\varepsilon$-DP.

**Theorem 2.6** (Basic composition (Dwork et al., 2006; Dwork & Lei, 2009)). *Consider algorithm $\mathcal{A}$ running $t$ (possibly adaptive) algorithms. If the $i$-th algorithm is $(\varepsilon_i, \delta_i)$-DP, for $\varepsilon_i, \delta_i \geq 0$, then $\mathcal{A}$ is $(\sum \varepsilon_i, \sum \delta_i)$-DP.*

## 3. Private Cut Sparsifier

In this section, we use our DP expander decomposition result stated in Theorem 3.1 (which we prove in Section 4) to describe our algorithm for private cut approximation.

**Theorem 3.1.** *Let $0 < \varepsilon, \delta < 1/2$ be parameters. For any $n$, there exists a parameter $\Psi(n) = n^{0.5+o(1)} \cdot \text{polylog}(1/\delta)/\varepsilon$, so that given an $n$-node non-negative weighted graph $G = (V, E, w)$ and a parameter $\psi \geq \Psi(n)$, there exists a polynomial time $(\varepsilon, \delta)$-DP algorithm that outputs a partition of $V = V_1 \sqcup \cdots \sqcup V_k$, such that with probability $1 - O(1/n^8)$,*

- *For every $1 \leq i \leq k$, the sparsity of $G[V_i]$ is at least $\psi$.*

- *The total weights of inter-component edges is $\psi \cdot n^{1+o(1)}$.*

Thus we get the following theorem.

**Theorem 3.2.** *Let $0 < \varepsilon, \delta < 1/2, 0 < \alpha < 1$ be parameters. Given any non-negative weighted $n$-node graph $G = (V, E, w)$, there is an $(\varepsilon, \delta)$-DP algorithm running in polynomial time that outputs a synthetic graph $\widetilde{G}$ on the same vertex set $V$ with non-negative edge weights, so that with high probability, for every $C \subseteq V$,*

$$(1 - \alpha)w_G(C) - \Delta \leq w_{\widetilde{G}}(C) \leq (1 + \alpha)w_G(C) + \Delta,$$

*for some*

$$\Delta = \frac{n^{1.25+o(1)} \cdot \text{polylog} \frac{1}{\delta}}{\alpha^{0.5} \cdot \varepsilon}. \qquad (1)$$

*Proof.* The algorithm is described in Algorithm 1.

---

**Algorithm 1** Differentially Private algorithm for preserving all cuts in a graph

---

1: **Input:** An $n$-node graph $G = (V, E, w)$; DP parameters $\varepsilon, \delta$
2: Compute $V_1, V_2, \ldots, V_k$ by invoking Theorem 3.1 on $G$ with parameters $\varepsilon/3, \delta/3$, and $\psi = O(\Psi(n)/\alpha \cdot \text{polylog}(n/\delta)) = n^{0.5+o(1)} \cdot \text{polylog}(1/\delta)/(\alpha\varepsilon)$.
3: Let $E_{\text{sparse}}$ be the edges of $E$ that are **not** within any $V_i$.
4: Let $\widetilde{G}_{\text{sparse}}$ be obtained by applying Theorem 2.3 with parameter $(\varepsilon/3, \delta/3)$ to $G_{\text{sparse}} = (V, E_{\text{sparse}}, w)$.
5: Let $\widetilde{G}_i$ be obtained by applying Theorem 2.2 on $G[V_i]$ with privacy parameter $(\varepsilon/3, \delta/3)$ and $\lambda = 1/n^{10}$.
6: **return** $\widetilde{G}$ whose edge weights are the corresponding edge weights from $\widetilde{G}_{\text{sparse}}$ and $\{\widetilde{G}_i\}_{i=1}^k$.

---

### 3.1. Privacy Guarantee

The only places where Algorithm 1 uses edge weights are via Theorem 3.1, Theorem 2.3 and Theorem 2.2. Let the edge difference between two inputs be on $e = \{u, w\}$. Note that there is only one invocation of Theorem 3.1 and one invocation of Theorem 2.3. To see the impact of Theorem 2.2 on privacy, consider the partitioning $V_1, \ldots, V_k$ of Theorem 3.1, which we assume is the same in both neighboring graphs. If $e$ is between two partitions, then the subgraph on each $V_i$ in both neighboring graphs is the same and the invocations of Theorem 2.2 yield the same output without losing any privacy. If $e$ is in some partition $V_i$, then only one invocation of Theorem 2.2 uses the privacy budget. Hence in total, at most three invocations of Theorem 3.1, Theorem 2.3 and Theorem 2.2 use privacy budget, and since each of these invocations is $(\varepsilon/3, \delta/3)$-DP, overall the algorithm is $(\varepsilon, \delta)$-DP by basic composition.

### 3.2. Success Probability

Several favorable events must occur so our algorithm yields the desired error bounds. First, we need the invocation of Theorem 3.1 and Theorem 2.3 to succeed. By union bound, these happen with high probability. Second, we need all invocations to Theorem 2.2 to succeed. Since there are at most $O(n)$ invocations, by union bound, the probability that any one of the invocations fails is $O(n\lambda) = O(1/n^9)$.

By union bound, the probability that none of these failure events occur is $1 - 1/\text{poly}(n)$. In the rest of our analysis, we condition on the good events occurring.

### 3.3. Additive Error Upper Bound

Here, we analyze the additive error of Algorithm 1. Let $C \subseteq V$ be an arbitrary subset. We aim to prove an upper bound $B$ on the additive error while allowing a $(1 + \alpha)$-

factor multiplicative error. That is, we want to show

$$(1-\alpha)w_G(C) - B \leq w_{\widetilde{G}}(C) \leq (1+\alpha)w_G(C) + B.$$

Hence, it suffices to show the following: $\left|w_{\widetilde{G}}(C) - w_G(C)\right| - \alpha \cdot w_G(C) \leq B$. To derive an upper bound on $\left|w_{\widetilde{G}}(C) - w_G(C)\right| - \alpha \cdot w_G(C)$, define $C_i \overset{\text{def}}{=} C \cap V_i$ for $1 \leq i \leq k$, where $V_i$ is obtained by Theorem 3.1. Then,

$$\begin{aligned}
&\left|w_{\widetilde{G}}(C) - w_G(C)\right| - \alpha \cdot w_G(C) \\
&\leq \left(\left|w_{\widetilde{G}_{\text{sparse}}}(C) - w_{G_{\text{sparse}}}(C)\right| - \alpha \cdot w_{G_{\text{sparse}}}(C)\right) \\
&+ \sum_{i=1}^{k}\left(\left|w_{\widetilde{G}_i}(C_i) - w_{G_i}(C_i)\right| - \alpha \cdot w_{G_i}(C_i)\right).
\end{aligned}$$

We now analyze each of the two terms separately.

**Additive error of $G_{\text{sparse}}$.** Since

$$\begin{aligned}
&\left|w_{\widetilde{G}_{\text{sparse}}}(C) - w_{G_{\text{sparse}}}(C)\right| - \alpha \cdot w_{G_{\text{sparse}}}(C) \\
&\leq \left|w_{\widetilde{G}_{\text{sparse}}}(C) - w_{G_{\text{sparse}}}(C)\right|,
\end{aligned}$$

we focus on upper-bounding the latter term.

By Theorem 3.1, the total edge weights of edges crossing two diffrent $V_i$ and $V_i$, $w(E_{\text{sparse}})$, are $\psi \cdot n^{1+o(1)} = n^{1.5+o(1)} \operatorname{polylog}(1/\delta)/(\alpha\varepsilon)$.

Therefore, by Theorem 2.3, we conclude that

$$\begin{aligned}
\left|w_{\widetilde{G}_{\text{sparse}}}(C) - w_{G_{\text{sparse}}}(C)\right| &\leq \widetilde{O}\left(\sqrt{\frac{w(E_{\text{sparse}})n}{\varepsilon}}\right) \\
&= \frac{n^{1.25+o(1)} \cdot \operatorname{polylog} \frac{1}{\delta}}{\alpha^{0.5} \cdot \varepsilon}. \quad (2)
\end{aligned}$$

**Additive error of $G_i$.** Let $n_i$ be the number of vertices in $V_i$. As a reminder, we defined $C_i \overset{\text{def}}{=} C \cap V_i$, where $V_i$ is obtained by Theorem 3.1. Let $D_i$ be the smaller one of $C_i$ and $V_i \setminus C_i$.

By Theorem 2.2, we have $\left|w_{\widetilde{G}_i}(D_i) - w_{G_i}(D_i)\right| \leq \frac{|D_i|\sqrt{n}\operatorname{polylog}(n/\delta)}{\varepsilon}$. By Theorem 3.1, we have $\phi_{G_i}(D_i) \geq \psi$, which implies $w_{G_i}(D_i) \geq |D_i|\psi = \frac{|D_i|n^{0.5+o(1)} \cdot \operatorname{polylog}(1/\delta)}{\alpha\varepsilon}$. Hence, we have

$$\begin{aligned}
&\left|w_{\widetilde{G}_i}(C_i) - w_{G_i}(C_i)\right| - \alpha \cdot w_{G_i}(C_i) \\
&= \left|w_{\widetilde{G}_i}(D_i) - w_{G_i}(D_i)\right| - \alpha \cdot w_{G_i}(D_i) \leq \quad 0, \quad (3)
\end{aligned}$$

if we set the $\operatorname{polylog}(n/\delta)$ factor in $\psi$ large enough.

**Putting everything together.** By combining all the cases, i.e., Equations (2) and (3), the final upper-bound on the additive error can be written as $\frac{n^{1.25+o(1)} \cdot \operatorname{polylog}\frac{1}{\delta}}{\alpha^{0.5}\cdot\varepsilon}$. This concludes the proof.

$\square$

We are now ready to prove Theorem 1.1. The difference between Theorem 3.2 and Theorem 1.1 is that the latter outputs a *sparse* graph with a near linear number of edges. The proof of Theorem 1.1 follows directly from Theorem 3.2: we obtain a synthetic private graph $\widetilde{G}$. Then, we run a vanilla non-private cut-sparsification algorithm (which is private via post-processing), such as (Benczúr & Karger, 1996), on $\widetilde{G}$, giving us a spare synthetic graph. Given a non-negative weighted graph and a parameter $\gamma$, the algorithm from (Benczúr & Karger, 1996) outputs a sparse graph with $\widetilde{O}(n/\gamma^2)$ edges in polynomial time where all cuts are approximated with a $(1+\gamma)$-multiplicative factor.

*Proof of Theorem 1.1.* Let $\gamma' = \gamma/100$. We run Theorem 3.2 to compute a synthetic graph $\widetilde{G}$ with $\alpha = \gamma'$, and then run (Benczúr & Karger, 1996)'s algorithm with parameter $\gamma'$ on $\widetilde{G}$ to find $H$. Let $\Delta = \frac{n^{1.25+o(1)}\operatorname{polylog}(1/\delta)}{\varepsilon\gamma^{0.5}}$ be the additive error guaranteed by Theorem 3.2. By the guarantee of (Benczúr & Karger, 1996), the number of edges of $H$ is $\widetilde{O}(n/\gamma^2)$, and for any $S \subseteq V$, we have

$$\begin{aligned}
&\left|w_G(S) - w_H(S)\right| \\
&\leq \left|w_G(S) - w_{\widetilde{G}}(S)\right| + \left|w_{\widetilde{G}}(S) - w_H(S)\right| \\
&\leq (\gamma' w_G(S) + \Delta) + (\gamma' \cdot w_{\widetilde{G}}(S)) \\
&\leq (\gamma' w_G(S) + \Delta) + (\gamma' \cdot ((1+\gamma') \cdot w_G(S) + \Delta)) \\
&\leq \gamma \cdot w_G(S) + O(\Delta). \quad\quad \square
\end{aligned}$$

## 4. Private Expander Decomposition

In this section we describe how to design a private expander decomposition algorithm. To that end, we first state a result that directly follows from prior work.

**Theorem 4.1** (Theorem 2.14 in (Li & Saranurak, 2021))**.** *Given a positive weighted $n$-node graph $G = (V, E, w)$ where the ratio between the largest edge weight and the smallest edge weight is bounded by $U$, and parameter $\psi > 0$, there is a deterministic $\operatorname{poly}(n, \log U)$-time algorithm that finds a cut $(S, V \setminus S)$ with $|S| \leq |V \setminus S|$ such that*

- $w(S) \leq \psi|S|$;

- *For any cut $(S', V \setminus S')$ with $|S'| \leq |V \setminus S'|$ and $w(S') \leq \psi|S'|/d_{exp}$, we have that $|S| \geq |S'|/d_{size}$ for some parameters $1 \leq d_{exp}(n) = \log^{O(1)}(n), 1 \leq d_{size}(n) = O(1)$.*

Note that Theorem 4.1 essentially gives a bicriteria-approximation of the largest set with sparsity at most $\psi$, where the approximation factors are denoted by $d_{\mathrm{exp}}$ and $d_{\mathrm{size}}$. In other words, the size of the set $S$ output by Theorem 4.1 is a $d_{\mathrm{size}}$-approximation of the largest set with sparsity at most $\psi/d_{\mathrm{exp}}$.

To obtain Theorem 4.1, we set the demand vector $\mathbf{d}$ to be the all-1 vector, and $r = 1$ in Theorem 2.14 in (Li & Saranurak, 2021). See also other versions of the results in, e.g., (Nanongkai & Saranurak, 2017; Wulff-Nilsen, 2017; Chuzhoy et al., 2020).

Note that Theorem 4.1 may return $S = \emptyset$, in which case it certifies that $\phi(S') > \psi/d_{\mathrm{exp}}$ for all cuts $(S', V \setminus S')$ with $0 < |S'| < n$. By combining Theorem 2.2 and Theorem 4.1, we can obtain a private version of Theorem 4.1 in a black-box way:

**Theorem 4.2.** *Let $\varepsilon, \delta, \lambda \in (0, 1/2)$ be parameters. For any $n$, there exists $\Psi(n) = \widetilde{\Omega}(\log(1/\lambda)\sqrt{n}/\varepsilon)$, so that given a non-negative weighted $n$-node graph $G = (V, E, w)$ and parameter $\psi \geq \Psi(n)$, there is a polynomial time $(\varepsilon, \delta)$-DP algorithm that finds a cut $(S, V \setminus S)$ with $|S| \leq |V \setminus S|$ such that with probability $1 - \lambda$ it holds:*

- $w(S) \leq \psi|S|$;

- *For any cut $(S', V \setminus S')$ with $|S'| \leq |V \setminus S'|$ and $w(S') \leq \psi|S'|/c_{exp}$, we have that $|S| \geq |S'|/c_{size}$ for some parameters $c_{exp}(n) = 2d_{exp}(n) = \log^{O(1)}(n), c_{size}(n) = d_{size}(n) = O(1)$.*

*Proof.* We first apply Theorem 2.2 with parameters $\varepsilon, \delta, \lambda$ to the input graph to obtain a graph $\widetilde{G}$. With probability $1 - \lambda$, we have $\left|w_G(S) - w_{\widetilde{G}}(S)\right| \leq \Delta|S|$ for $\Delta = O(\log(1/\lambda)\sqrt{n \log n \log(1/\delta)}/\varepsilon)$, for all $S \subseteq V$. We assume this holds in the remainder of the proof. Let $\Psi(n) = 10\Delta d_{\mathrm{exp}}(n)$.

If some edge in $\widetilde{G}$ has weight smaller than $\psi/10n$, we simply remove this edge. Furthermore, if some edge in $\widetilde{G}$ has weight greater than $\psi n$, we set its weight to $\psi n$. We call the updated graph $\widetilde{G}'$. This way, the ratio between the largest edge weight and the smallest edge weight in $\widetilde{G}'$ is bounded by $U = O(n^2)$.

Then we run Theorem 4.1 on $\widetilde{G}'$ with parameter $0.8\psi$ to obtain a cut $(S, V \setminus S)$ with $|S| \leq |V \setminus S|$. The running time is hence $\mathrm{poly}(n, \log U) = \mathrm{poly}(n)$.

Theorem 4.1 guarantees that (1) $w_{\widetilde{G}'}(S) \leq 0.8\psi|S|$ and (2) For any cut $(S', V \setminus S')$ with $|S'| \leq |V \setminus S'|$ and $w_{\widetilde{G}'}(S') \leq 0.8\psi|S'|/d_{\mathrm{exp}}$, we have $|S| \geq |S'|/d_{\mathrm{size}} = |S'|/c_{\mathrm{size}}$. Then,

- Because $w_{\widetilde{G}'}(S) \leq 0.8\psi|S|$, there is no edge in the cut with weight $\psi n$. Hence, the difference between $w_{\widetilde{G}'}(S)$

and $w_{\widetilde{G}}(S)$ is only caused by edges in $\widetilde{G}$ with weight smaller than $\psi/10n$. Therefore, $w_{\widetilde{G}}(S) \leq w_{\widetilde{G}'}(S) + (\psi/10n) \cdot |S|(n - |S|) \leq 0.9\psi|S|$. Furthermore,

$$w_G(S) \leq w_{\widetilde{G}}(S) + \left|w_G(S) - w_{\widetilde{G}}(S)\right|$$
$$\leq 0.9\psi|S| + \Delta|S| \leq \psi|S|,$$

where the final step holds because $\psi \geq \Psi(n) \geq 10\Delta d_{\mathrm{exp}}$.

- For any cut $(S', V \setminus S')$ with $|S'| \leq |V \setminus S'|$ and $w_G(S') \leq \psi|S'|/c_{\mathrm{exp}}$, we have

$$w_{\widetilde{G}}(S') \leq w_G(S') + \left|w_G(S') - w_{\widetilde{G}}(S')\right|$$
$$\leq \psi|S'|/c_{\mathrm{exp}} + \Delta|S'| = \psi|S'|/2d_{\mathrm{exp}} + \Delta|S'|.$$

The above can be upper bounded by $0.6\psi|S'|/d_{\mathrm{exp}}$ because $\psi \geq \Psi(n) \geq 10\Delta d_{\mathrm{exp}}$. Furthermore, $w_{\widetilde{G}'}(S') \leq w_{\widetilde{G}}(S')$ because we only decrease edge weights or remove edges in $\widetilde{G}'$ compared to $\widetilde{G}$. Therefore,

$$w_{\widetilde{G}'}(S') \leq 0.6|S'|/d_{\mathrm{exp}} \leq 0.8|S'|/d_{\mathrm{exp}},$$

and we can apply the second guarantee of Theorem 4.1 to obtain $|S| \geq |S'|/c_{\mathrm{size}}$ as desired.

The privacy guarantee follows because Theorem 2.2 is $(\varepsilon, \delta)$-DP, and the remainder of the algorithm is post-processing on the private output of Theorem 2.2. $\qquad\square$

Using Theorem 4.2, one could obtain a private expander decomposition algorithm using the method in (Nanongkai & Saranurak, 2017) for a given graph $G$ and a sparsity parameter $\psi$. To describe the algorithm, we define a few parameters. If $n$ is the number of vertices in the input graph, then $\bar{c}_{\mathrm{size}} = c_{\mathrm{size}}(n)$, $\bar{c}_{\mathrm{exp}} = c_{\mathrm{exp}}(n)$, and $\sigma = \sqrt{\log \bar{c}_{\mathrm{exp}}/\log n}$. $\bar{s}_1, \ldots, \bar{s}_L$ are parameters where $\bar{s}_1 = n/2 + 1$, $\bar{s}_i = \bar{s}_{i-1}/n^\sigma$ for $i > 1$, and $L$ is the smallest integer where $\bar{s}_L \leq 1$ (these imply $L = O(1/\sigma)$). Also, let $\psi_i = \psi \cdot (\bar{c}_{\mathrm{exp}})^{L-i+1}$ for $1 \leq i \leq L$. Let $\mathcal{A}(H, \psi)$ be the returned result of applying Theorem 4.2 on graph $H$ with DP parameters $\varepsilon' = \varepsilon/(L\bar{c}_{\mathrm{size}}n^\sigma \log n), \delta' = \delta/(L\bar{c}_{\mathrm{size}}n^\sigma \log n)$, probability parameter $\lambda = 1/n^{10}$, and sparsity parameter $\psi$.

The algorithm is described in Algorithm 2. Note that compared to (Nanongkai & Saranurak, 2017), our description of the algorithm omits one variable $I$ to the recursive calls, as it was not used by the algorithm and only used for analysis in (Nanongkai & Saranurak, 2017).

The following two lemmas summarize some of the results proved in (Nanongkai & Saranurak, 2017) regarding Algorithm 2. Lemma 4.3 shows that Algorithm 2 satisfies the properties we want for our expander decomposition, and Lemma 4.4 bounds the recursion depth of Algorithm 2 which is used in proving privacy. Even though in

**Algorithm 2** The expander decomposition algorithm in (Nanongkai & Saranurak, 2017). $\mathcal{A}(H, \psi)$ is the returned result of applying Theorem 4.2 on graph $H$ with DP parameters $\varepsilon' = \varepsilon/(L\bar{c}_{\text{size}}n^\sigma \log n)$, $\delta' = \delta/(L\bar{c}_{\text{size}}n^\sigma \log n)$, probability parameter $\lambda = 1/n^{10}$, and sparsity parameter $\psi$.

1: **Input:** A graph $H = (V, E, w)$; a level parameter $\ell$
2: $S \leftarrow \mathcal{A}(H, \psi_\ell)$
3: **if** $H$ is a singleton or $S = \emptyset$ **then**
4:     **return** $H$
5: **else**
6:     **if** $|S| \geq \bar{s}_{\ell+1}/\bar{c}_{\text{size}}$ **then**
7:         Recurse on $(H[S], 1)$ and $(H[V \setminus S], \ell)$, and return union of the results
8:     **else**
9:         Recurse on $(H, \ell + 1)$ and return the result.
10:     **end if**
11: **end if**

(Nanongkai & Saranurak, 2017) their whole algorithm is stated to work for unweighted graphs, these two lemmas still work for weighted graphs with little to no change of their proofs. The only slight change needed is for the second bullet point of Lemma 4.3, where (Nanongkai & Saranurak, 2017) showed an upper bound on the number of edges, but the same method can be extended to bound the total weight of edges for weighted graphs in a straighforward manner.

**Lemma 4.3** ((Nanongkai & Saranurak, 2017)). *If $\mathcal{A}$ always correctly returns a set $S$ that satisifes the guarantees in Theorem 4.2, then Algorithm 2 on input $(G = (V, E, w), 1)$ for an $n$-node non-negative weighted graph $G$ returns a partition of $V = V_1 \sqcup \cdots \sqcup V_k$, such that*

- *For every $1 \leq i \leq k$, the sparsity of $G[V_i]$ is at least $\psi$.*

- *The total weights of edges between different parts $V_i$ and $V_j$ for $i \neq j$ is $O(\psi_1 n \log n) = \psi \cdot n^{1+o(1)}$.*

**Lemma 4.4** ((Nanongkai & Saranurak, 2017)). *The recursion depth of Algorithm 2 on input $(G, 1)$ is at most $L\bar{c}_{size}n^\sigma \log n = n^{o(1)}$.*

Now we are ready to prove Theorem 3.1

*Proof of Theorem 3.1.* The algorithm is to simply call Algorithm 2 with input $(G, 1)$ and the value of $\Psi(n)$ is the same as that from Theorem 4.2.

In order to apply Lemma 4.3, we need to show that all invocations of algorithm $\mathcal{A}$ are correct with probability $1 - O(1/n^8)$. On each recursion level, the set of vertices in all the graphs $H$ in the recursive calls are disjoint, so the total number of recursive calls on each recursion level is $O(n)$. As the recursion depth is $n^{o(1)}$ by Lemma 4.4, the

total number of recursive calls, and hence the total number of invocations of $\mathcal{A}$, is $n^{1+o(1)}$. By union bound, the overall success probability is $\geq 1 - \lambda \cdot n^{1+o(1)} = 1 - O(1/n^8)$.

We also need to guarantee that all invocations of Theorem 4.2 have sparsity parameter $> \Psi(|H|)$ as required. This holds because the smallest sparsity parameter we use is $\psi_L = \psi > \Psi(n) \geq \Psi(|H|)$.

Hence, all invocations of $\mathcal{A}$ are correct with probability $1 - O(1/n^8)$, so the correctness follows from Lemma 4.3.

Finally, we analyze the privacy guarantee of Algorithm 2. Let the edge difference between two inputs be at $e = \{u, w\}$. Observe that when Algorithm 2 recurses, the edge $e$ affects at most one of the recursive calls. Therefore, the number of recursive calls affected by $e$ is upper-bounded by the recursion depth, which is bounded by $L\bar{c}_{\text{size}}n^\sigma \log n$. The only part in Algorithm 2 that uses edge weights is via $\mathcal{A}$, which is $(\varepsilon/L\bar{c}_{\text{size}}n^\sigma \log n, \delta/L\bar{c}_{\text{size}}n^\sigma \log n)$-DP. Hence, Algorithm 2 is $(\varepsilon, \delta)$-DP by basic composition. $\square$

## 5. Applications

We highlight some downstream applications of our main result beyond the improved error bound for private all-cuts. These applications include many of those in prior DP graph cut works, e.g., (Arora & Upadhyay, 2019), except we use our improved bound from Theorem 1.1 as needed. We present our max-cut application in the main body and discuss our other applications to the maximum-bisection, max-$k$-cut, and minimum-bisection problems in Appendix B.

**Max-Cut.** In the Max-Cut problem, we are given a weighted graph $G$ (with non-negative weights), and the goal is to output a subset $S \subseteq V$ maximizing $w_G(S)$. Goemans and Williamson gave a polynomial time algorithm for computing a $\zeta_{GW} - \eta$ approximation for any $\eta > 0$ where $\zeta_{GW} \approx 0.87856$ (Goemans & Williamson, 1995). Furthermore, it is known that assuming the Unique-Games conjecture, it is NP-Hard to approximate Max-Cut to any factor better than $\zeta_{GW} + \rho$ for any $\rho > 0$ (Khot et al., 2007).

With respect to DP, the prior state-of-the-art algorithms obtain the same multiplicative approximation as above, with additional additive error coming from cut-sparsification. In particular, it is known that for any fixed $\eta > 0$, there is a polynomial time algorithm satisfying $\varepsilon$-DP with respect to edge-neighboring graphs, which on any non-negative weighted input graph $G = (V, E)$, outputs a subset $S \subseteq V$ satisfying $w_G(S) \geq (\zeta_{GW} - \eta) \max_{S' \subseteq V} w_G(S') - \widetilde{O}\left(\frac{n^{1.5}}{\varepsilon}\right)$, meeting the same $\widetilde{O}(n^{1.5})$ factor from cut-sparsification (Gupta et al., 2012).

We note that even if approximate-DP is allowed, there are no algorithms obtaining better than $\widetilde{O}(n^{1.5})$ error, al-

though we remark that (Arora & Upadhyay, 2019) obtain a more refined guarantee with error of the form $\widetilde{O}(|S'|\sqrt{n}\operatorname{polylog}(1/\delta)/\varepsilon)$ where $|S'|$ is the cardinality of the optimal set maximizing the cut value. However, in the worst case, this bound is still $\widetilde{\Omega}(n^{1.5})$. Using our synthetic graph leads to improved guarantees, a corollary of our main theorem, with a full proof in Appendix B.

**Corollary 5.1** (Corollary of our Main Theorem 3.2). *For any fixed $\eta > 0$, there exists a polynomial time $(\varepsilon, \delta)$-DP algorithm that on any non-negative weighted $n$-node graph $G = (V, E, w)$ outputs a subset $S \subseteq V$ satisfying (with high probability) that $w_G(S)$ is at least*

$$(\zeta_{GW} - \eta) \max_{S' \subseteq V} w_G(S') - O(\Delta).$$

## 6. Conclusion

Given a graph $G$, we develop an algorithm that, in a DP manner, computes a synthetic graph whose cut values are close to those in $G$. On $n$-node graphs with more than $n^{1.5}$ edges, our algorithm imposes significantly lower additive error than the prior work. We obtain this result by decomposing the input graph into sparse and dense regions. As a byproduct, we show how to design a DP algorithm for expander decomposition. One interesting open question is whether private expander decomposition has applications beyond all cuts.

Our work leaves a few intriguing questions. Perhaps the most natural one is whether the additive error can be reduced beyond $n^{1.25}$ while still allowing for a multiplicative approximation. Interestingly, an improved additive error (at the cost of allowing multiplicative errors) for graphs with roughly $n^{1.5}$ edges would directly imply improvements for all graphs by plugging it into our method. This yields another open question: What is the range of the number of edges $m \gg n$ for which the $\sqrt{mn}$ additive error can be polynomially improved while allowing multiplicative errors?

## Acknowledgements

A. Aamand was supported by the VILLUM Foundation grant 54451. J. Chen was supported by an NSF Graduate Research Fellowship under Grant No. 17453. S. Mitrović was supported by the NSF Early Career Program No. 2340048. Y. Xu was supported by HDR TRIPODS Phase II grant 2217058.

## Impact Statement

This paper presents work whose goal is to advance algorithmic tools applied in the field of Machine Learning. There are many potential societal consequences of our work, none of which we feel must be specifically highlighted here. Our work is theoretical in nature.

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

# A. Proof of Theorem 2.3

*Proof of Theorem 2.3.* In order to prove the theorem, we first introduce the notion of cut norm. For any real $n$ by $n$ matrix $A$, the cut norm of $A$ is defined as

$$\|A\|_C = \max_{I,J \subseteq [n]} \left| \sum_{i \in I} \sum_{j \in J} A_{i,j} \right|.$$

Given any matrix $A$, there is a polynomial time algorithm that outputs an estimate $\widetilde{\|A\|_C}$ such that $0.56\|A\|_C \le \widetilde{\|A\|_C} \le \|A\|_C$ (Alon & Naor, 2006).

For an undirected graph $G$, let $W_G$ denote its weighted adjacency matrix. Given a non-negative weighted graph $G$, (Eliáš et al., 2020) showed an $(\varepsilon, \delta)$-DP polynomial time algorithm that outputs a non-negative weighted graphs $\widetilde{G}$ such that

$$\mathbb{E}\left[\|W_G - W_{\widetilde{G}}\|_C\right] \le O\left(\sqrt{\frac{w(E)n}{\varepsilon}} \log^2\left(\frac{n}{\delta}\right)\right).$$

Let $L$ be $O(\log n)$ with a sufficiently large constant hidden in front of $O$. We run (Eliáš et al., 2020)'s algorithm with parameters $(\varepsilon/L, \delta/L)$ independently for $L$ times to obtain $\widetilde{G}_1, \ldots, \widetilde{G}_L$. By Markov's inequality, for each $\widetilde{G}_i$, $\mathbb{E}\left[\|W_G - W_{\widetilde{G}_i}\|_C\right] \le \Delta$ for some $\Delta = O\left(\sqrt{\frac{w(E)n \log n}{\varepsilon}} \log^2\left(\frac{n \log n}{\delta}\right)\right)$ with probability at least $2/3$.

The algorithm then proceeds to compute an approximation $\tilde{d}_{i,j}$ of $\|\widetilde{G}_i - \widetilde{G}_j\|_C$ for every $i, j \in [L]$ using (Alon & Naor, 2006)'s algorithm. The algorithm then outputs an arbitrary $\widetilde{G}_i$ as the result where $\tilde{d}_{i,j} \le 2\Delta$ for more than half of the indices $j \in [L]$ (break ties arbitrarily; if there is no such $i$ satisfying the requirement, the algorithm may output an empty graph).

First of all, this graph is $(\varepsilon, \delta)$-DP by basic composition.

Next, we show the correctness of the algorithm with high probability. By standard concentration bounds, $\mathbb{E}\left[\|W_G - W_{\widetilde{G}_i}\|_C\right] \le \Delta$ holds for more than half of the synthetic graphs $\widetilde{G}_i$ with high probability. We next condition on this event.

Let $I \subseteq [L]$ with $|I| > L/2$ be the set of indices $i$ with $\mathbb{E}\left[\|W_G - W_{\widetilde{G}_i}\|_C\right] \le \Delta$. For any $i, j \in I$,

$$\tilde{d}_{i,j} \le \|W_{\widetilde{G}_i} - W_{\widetilde{G}_j}\|_C \le \|W_{\widetilde{G}_i} - W_G\|_C + \|W_G - W_{\widetilde{G}_j}\|_C \le 2\Delta.$$

Hence, for any $i \in I$, the number of $j$ where $\tilde{d}_{i,j} \le 2\Delta$ is more than $L/2$, so the algorithm will not output an empty graph.

Now let $\widetilde{G}_i$ be the graph output by the algorithm. Since there are more than $L/2$ indices $j$ with $\tilde{d}_{i,j} \le 2\Delta$, and $|I| > L/2$, there must exist some $j \in I$ where $\tilde{d}_{i,j} \le 2\Delta$. Then

$$\|W_{\widetilde{G}_i} - W_G\|_C \le \|W_{\widetilde{G}_i} - W_{\widetilde{G}_j}\|_C + \|W_{\widetilde{G}_j} - W_G\|_C \le \frac{1}{0.56}\tilde{d}_{i,j} + \Delta = O(\Delta).$$

For any cut $(S, V \setminus S)$, we have

$$\left| w_{\widetilde{G}_i}(S) - w_G(S) \right| = \left| \sum_{u \in S} \sum_{v \in V \setminus S} \left( W_{\widetilde{G}_i} - W_G \right)_{u,v} \right| \le \|W_{\widetilde{G}_i} - W_G\|_C = O(\Delta),$$

as desired.

$\square$

# B. Omitted Applications

We discuss additional applications of our main result to the maximum bisection, max-$k$-cut, and minimum bisection problems. We also give the full proof of Corollary 5.1 for our max-cut application.

**Max-Cut.**

*Proof of Corollary 5.1.* The algorithm is straightforward. We compute our private synthetic graph $\widetilde{G}$ on the input graph $G$, letting $\alpha = \eta/100$ in Theorem 3.2. Then, we run the best off-the-shelf approximation algorithm for max-cut on $\widetilde{G}$. This yields a solution in $\widetilde{G}$ with $\zeta_{GW} - \eta/100$ multiplicative error. Note that privacy is not affected since the latter step is post-processing. In our private synthetic graph, *every* cut is approximated up to additive error $\Delta$ as defined in Equation 1. Letting $S$ be the subset of $V$ that we output, we have

$$
\begin{aligned}
w_G(S) &\geq \frac{1}{1+\alpha} w_{\widetilde{G}}(S) - \Delta \\
&\geq \frac{\zeta_{GW} - \eta/100}{1+\alpha} \max_{S' \subseteq V} w_{\widetilde{G}}(S') - \Delta \\
&\geq \frac{\zeta_{GW} - \eta/100}{1+\alpha} \max_{S' \subseteq V} ((1-\alpha)w_G(S') - \Delta) - \Delta \\
&\geq \frac{(\zeta_{GW} - \eta/100)(1-\alpha)}{1+\alpha} \max_{S' \subseteq V} w_G(S') - O(\Delta) \\
&\geq (\zeta_{GW} - \eta) \max_{S' \subseteq V} w_G(S') - O(\Delta),
\end{aligned}
$$

as desired. $\qquad\square$

**Maximum-Bisection and Max-$k$-Cut.** The Maximum-Bisection problem is a well-studied variant of Max-Cut with *balance* constraints. More precisely, given a weighted graph $G$, we seek to find a cut $S$ satisfying $|S| = n/2$ – we can assume $n$ is even by adding an isolated node – such that $w_G(S)$ is maximized (Frieze & Jerrum, 1997).

Max-$k$-Cut is an alternate way to generalize Max-Cut where we seek to partition $V$ into $k$ pieces, maximizing the total weight of edges across distinct pieces (Frieze & Jerrum, 1997). Since these problems generalize Max-Cut, both problems cannot be solved exactly in polynomial time unless $P = NP$. Furthermore, similar to Max-Cut, it is known that Maximum-Bisection cannot be approximated better than $\zeta_{GW} + \eta$ for any $\eta > 0$ (Khot et al., 2007; Raghavendra & Tan, 2012), under the Unique-Games conjecture. For upper bounds, it is known that Maximum-Bisection can be approximated to a factor $\zeta_{MB} - \eta$ for any $\eta > 0$ where $\zeta_{MB} \approx 0.8776$ (note that this is not quite the same as $\zeta_{GW}$, although it is close) (Austrin et al., 2016), and Max-$k$-Cut can be approximated to a multiplicative factor $c_k > 1 - 1/k$ for any $k \geq 1$ (Frieze & Jerrum, 1997; de Klerk et al., 2004). We remark that better bounds are known for small values of $k$ (Frieze & Jerrum, 1997; de Klerk et al., 2004).

To obtain DP algorithms for these problems, we follow the same approach as for the previous applications. Namely, we invoke the appropriate non-private optimization algorithms on a private synthetic graph, resulting in an $\varepsilon$-DP algorithm with the same multiplicative factor approximation as non-private, and with additive error $\widetilde{O}(n^{1.5}/\varepsilon)$ in the Maximum-Bisection problem, and $\widetilde{O}(kn^{1.5}/\varepsilon)$ for Max-$k$-Cut. For example, this follows from the work of (Gupta et al., 2012).

Our main result yields improved bounds, as stated by the following claim. The proof is almost identical to that of Corollary 5.1, so we omit it.

**Corollary B.1** (Corollary of our Main Theorem 3.2)**.** *There exists a polynomial time $(\varepsilon, \delta)$-DP algorithm that on any non-negative weighted $n$-node graph $G = (V, E, w)$ outputs a subset $S \subseteq V, |S| = n/2$ satisfying (with high probability) that $w_G(S)$ is at least*

$$
(\zeta_{MB} - \eta) \max_{S' \subseteq V, |S'| = n/2} w_G(S') - \frac{n^{1.25 + o(1)} \cdot \text{polylog} \frac{1}{\delta}}{\varepsilon}.
$$

*Similarly, for any $k \geq 1$, there exists an $(\varepsilon, \delta)$-DP algorithm that runs in time $\text{poly}(n, \log\log(1/\delta), \log(1/\varepsilon))$ and partitions $V$ into $S_1, \cdots, S_k$ such that (with high probability) the total weight of the edges between distinct partitions is at least $(c_k - \eta)$ fraction of the optimal solution up to an additive error*

$$
\frac{kn^{1.25 + o(1)} \cdot \text{polylog} \frac{1}{\delta}}{\varepsilon}.
$$

**Minimum-Bisection.** This application is the natural *minimizing* variant of Maximum-Bisection. Recall that a bisection of a graph is a partition of its nodes into two equal-sized sets, so we want to minimize $w(S)$ subject to $|S| = n/2..$ The cost of

a bisection is the weighted cost of edges crossing the cut. The study of this problem can be traced back to almost five decades ago (Garey et al., 1974), and it is known that solving the problem exactly is NP-hard. The best-known polynomial-time algorithm achieves a $O(\log n)$-approximation (Räcke, 2008).

Concerning DP, prior work on DP all-cuts can again be applied straightforwardly (simply run the bisection algorithm of (Räcke, 2008) as post-processing), which gives an $\varepsilon$-DP algorithm computing the minimum-bisection with multiplicative approximation $O(\log(n))$ and additive error $\widetilde{O}(n^{1.5}/\varepsilon)$. Again, we note that even if approximate DP is allowed, previous algorithms cannot improve the additive error. Our main theorem gives the following corollary, which is obtained by again running the standard bisection algorithm of (Räcke, 2008) on the output of our main theorem as post-processing.

**Corollary B.2** (Corollary of our Main Theorem 3.2). *There is a polynomial time algorithm satisfying $(\varepsilon, \delta)$-DP, which on any nonnegative weighted $n$-node graph $G = (V, E, w)$ outputs a subset $S \subseteq V$ satisfying $|S| = n/2$ and $w_G(S)$ is (with high probability) at most*

$$O(\log(n)) \cdot \min_{S' \subseteq V, |S'| = n/2} w_G(S') + \frac{n^{1.25 + o(1)} \cdot \text{polylog} \frac{1}{\delta}}{\varepsilon}.$$

