# OpenReview forum: "Breaking the $n^{1.5}$ Additive Error Barrier for Private and Efficient Graph Sparsification via Private Expander Decomposition"
_ICML.cc/2025/Conference — ICML 2025 poster_

### Official Review · Reviewer_xtuG · 2025-03-11

**Overall Recommendation:** 4

**Summary:**

This paper focuses on designing differentially private algorithms for graph cut sparsification.

The previously best-known private and efficient cut sparsifiers on n-node graphs approximate each cut within $O(n^{1.5})$ additive error and $1+\gamma$ multiplicative error for any $\gamma>0$.
Exponential time algorithms can achieve an $O(n)$ additive error and $1+\gamma$ multiplicative error.

This work breaks the $n^{1.5}$ additive error barrier for private and efficient cut sparsification.
The authors present an $(\varepsilon, \delta)$-DP polynomial time algorithm that, given a non-negative weighted graph, outputs a private synthetic graph approximating all cuts with multiplicative error $1+\gamma$ and additive error $n^{1.25+o(1)}$ (ignoring dependencies on $\varepsilon, \delta, \gamma$).

The approach is based on a private algorithm for expander decomposition, a technique in graph algorithms.

---
## update after rebuttal
Sparsification is an important procedure in graph algorithms, and private implementations can lead to improved private graph algorithms. Given the nice theoretical improvement and the independent contribution of private expander decomposition, as confirmed in the rebuttal, I remain positive about the result.

**Claims And Evidence:**

All claims are theoretical and are supported by formal proofs.

**Essential References Not Discussed:**

N/A

**Experimental Designs Or Analyses:**

N/A

**Methods And Evaluation Criteria:**

N/A

**Other Comments Or Suggestions:**

There is a possible typo on line 365: all $S$ should be $S’$

Are there any prior applications of private expander decomposition? If not, I think this fact should be highlighted more.

**Other Strengths And Weaknesses:**

Strengths:
- The application of expander decomposition for graph DP appears to be novel and can likely find applications elsewhere.
- The overall algorithm is conceptually simple and is well-presented
- Private cut sparsification is an important problem with many downstream applications for DP graph algorithms, as shown by the authors.

Weaknesses:
- The key contribution of private expander decomposition consists of using a previous ($n^{1.5}$ additive error) DP cut approximator and running a non-private sparsest cut algorithm recursively. While the combination requires some cleverness, it does consist of stitching together existing results.
- While it is justified by lower bounds, an argument could be made that the additive error is large and is unlikely to yield a practical implementation.

That being said, the theoretical improvement is nice and would still be a great addition to ICML

**Questions For Authors:**

N/A

**Relation To Broader Scientific Literature:**

This work breaks the $n^{1.5}$ additive error barrier for private and efficient cut sparsification and achieves multiplicative error $1+\gamma $ and additive error $n^{1.25+o(1)}$. The previously best-known private and efficient cut sparsifiers on $n$-node graphs approximate each cut within $O(n^1.5)$ additive error and $1+\gamma$ multiplicative error.

**Theoretical Claims:**

I verified all proofs in the main body but not extremely thoroughly.

---

> ### Author Rebuttal · Authors · 2025-04-01
>
> Thank you for your valuable feedback!
>
> > While it is justified by lower bounds, an argument could be made that the additive error is large and is unlikely to yield a practical implementation.
>
> We are hopeful that for dense graphs, where cuts can be as large as $\Omega(n^2)$, our techniques could yield improved practical results over the prior $n^{1.5}$ error algorithms. However, our focus is on proving an improved asymptotic privacy-utility trade-off.
>
> > There is a possible typo on line 365: all S should be S’
>
> Thanks, we will update the typo.
>
> > Are there any prior applications of private expander decomposition? If not, I think this fact should be highlighted more.
>
> To the best of our knowledge, we are the first to study private expander decomposition. In our paper we mainly focused on cuts, and it is an interesting future direction to study further applications of our private expander procedure. We will make sure to highlight this in the paper.

---

### Official Review · Reviewer_qAzu · 2025-03-12

**Overall Recommendation:** 4

**Summary:**

This paper investigates $(\varepsilon, \delta)$ differentially private graph cut sparsification under edge-privacy.
More exactly, given a non-negative, undirected graph $G=(V, E, w)$ the goal is to output a non-negative, weighted, undirected graph $\tilde{G}$ that (1) approximates the value of all cuts in $G$ and (2) satisfies differential privacy.
Error is measured as $\lvert w(G) - w(\tilde{G}) \rvert \leq \gamma w(G) + \alpha$ for $\gamma \geq 0$, and we are primarily interested in dense input graphs.

For a purely additive error ($\gamma=0$), when an upper bound on the number of edges $m$ is known, error $\tilde{O}(\sqrt{mn})$ is possible (Elias et al., 2020, Liu et al., 2024).
For dense graphs more broadly $\alpha = O(n^{1.5})$ is always possible (Gupta et al., 2012) and there is a matching lower bound (Elias et al., 2020; Liu et al., 2024).
However, once a multiplicative error ($\gamma > 0$) is also allowed, this lower bound no longer applies.
Allowing for a multiplicative error, $\alpha = O(n\log n)$ is known (Elias et al., 2020) which is known to be near-optimal (Dalirrooyfard et al., 2023), but this algorithm requires exponential time.
The main question addressed in this paper is closing the gap between the $O(n^{1.5})$ and $\tilde{O}(n)$ additive error, while allowing $\gamma > 0$, for algorithms running in polytime.

This paper makes meaningful progress towards this by designing a polytime algorithm with dependence $n^{1.25 + o(1)}$ for their additive error.
The central idea in the paper is to implement a private expander decomposition (Theorem 3.1) and apply it to $G$ to yield partition of the vertices $V_1,\dots, V_k$.
Each of the induced subgraphs $G[V_1], \dots, G[V_k]$ will be dense, and the remaining edges not included in the subgraphs form a sparse graph on $V$, call it $G_{sparse}$.
Running existing private cut approximation algorithms for carefully chosen parameters on $G[V_1], \dots, G[V_k]$ and $G_{sparse}$, leveraging the density of the former and the sparsity of the latter, gives us private synthetic graphs $\tilde{G}[V_1], \dots, \tilde{G}[V_k], \tilde{G}_{sparse}$.
Returning the union of all of these edge weights gives a synthetic graph $\tilde{G}$, with non-negative edge-weights that preserves all cuts up to the aforementioned errors (Theorem 3.2).
To get a sparse synthetic graph with similar guarantees, the authors show that post-processing $\tilde{G}$ with a non-private cut-sparsification algorithm allows for recovering much the same guarantee (Theorem 1.1).
This last step, however, can be performed for all algorithms in Table 1 that achieve non-negative weights.

The authors go on to show that their algorithm can be used for downstream applications, including max-cut, where their results imply better polytime algorithms for these problems.

## update after rebuttal

I had no questions, but after taking all other rebuttals into account, I feel confident in keeping my score as is.

**Claims And Evidence:**

Yes.

**Essential References Not Discussed:**

Not that I am aware of.

**Experimental Designs Or Analyses:**

N/A.

**Methods And Evaluation Criteria:**

Yes.

**Other Comments Or Suggestions:**

N/A.

**Other Strengths And Weaknesses:**

Strengths:
1. The paper is well-written. The balance between intuition and technical detail is good.
2. The problem being studied is practically motivated, and I think the main algorithm is a meaningful contribution.
3. The idea for the main algorithm seems natural in hindsight, but requires some technical work (notably the private expander decomposition).
4. The private expander decomposition could see use for other problems in this domain.

Weaknesses:
Nothing comes to mind.

**Questions For Authors:**

I have no questions.

**Relation To Broader Scientific Literature:**

The results in this paper achieve state of the art for private cut sparsification in polytime.
The private expander decomposition in the paper is of independent interest.

**Theoretical Claims:**

Yes, I read through the proofs in the main part of the paper.

---

> ### Author Rebuttal · Authors · 2025-04-01
>
> Thank you for your valuable feedback!

---

### Official Review · Reviewer_E2qf · 2025-03-16

**Overall Recommendation:** 5

**Summary:**

The paper studies the problem of graph sparsification which preserves all cuts, a fundamental problem in graph algorithms, under differential privacy. This problem has been well studied but remains open. The paper makes significant improvements to prior work in terms of the additive approximation factor and running time, in particular beating the n^{1.5} bound, which held for a long time. The technical contributions are strong, and the presentation is quite good.

**Claims And Evidence:**

Yes

**Essential References Not Discussed:**

None

**Experimental Designs Or Analyses:**

N/A

**Methods And Evaluation Criteria:**

This is a theoretical paper

**Other Comments Or Suggestions:**

In definition 2.4, since non-edges are viewed as weight 0 edges, and (u, v) can be any pair of nodes, the edge set should be V^2? So E and E’ both should be V^2. This would be relevant for all the formal statements which mention E.
Do the results hold if the set of edges is fixed and only the weights of two existing edges differ by 1?
It might be helpful to explain the main theorems a bit more as part of the technical contribution in section 1.2.
Line 240: “outputs partition of” --> “outputs a partition of”
It would be useful to explain the notion of private expanders

**Other Strengths And Weaknesses:**

The paper is technically quite interesting and makes a significant improvement over
prior results, as the authors compare well. The presentation is generally quite nice.
The authors also show that their methods lead to improvements to other important problems such as max-cut and max-k-cut.

**Questions For Authors:**

Is it possible to get better approximation using your approach if you preserve only large cuts lower bounded by some threshold?
Would the techniques work for other notions of expansion?

**Relation To Broader Scientific Literature:**

Very good comparison. The paper makes significant advances over prior bounds as described in Table 1

**Theoretical Claims:**

Not very carefully, but generally seem right

---

> ### Author Rebuttal · Authors · 2025-04-01
>
> Thank you for your valuable feedback!
>
> > In definition 2.4, since non-edges are viewed as weight 0 edges, and (u, v) can be any pair of nodes, the edge set should be V^2? So E and E’ both should be V^2. This would be relevant for all the formal statements which mention E.
>
> Thank you. This is a good point, and we will update the text to address it.
>
> > Do the results hold if the set of edges is fixed and only the weights of two existing edges differ by 1?
>
> Yes, this is an easier setting (consider the non-edges to be zero-weight edges), so our results also hold.
>
> > It might be helpful to explain the main theorems a bit more as part of the technical contribution in section 1.2.
>
> Thank you. We will add more exposition in Section 1.2 in the final version.
>
> > It would be useful to explain the notion of private expanders
>
> By an expander partition, we intuitively mean a partition of the nodes into some number of parts such that the “connectivity inside” every part is high and the total number of edges in between the partitions is low. We use sparsity to formalize “connectivity” (see line 140, right column). We want to output such a decomposition which respects edge-neighboring privacy.
>
> > Is it possible to get better approximation using your approach if you preserve only large cuts lower bounded by some threshold?
>
> Our results directly imply that we obtain a multiplicative approximation for all cuts above $n^{1.25}$ (ignoring logarithmic and privacy factors). This improves upon prior work which can only guarantee a multiplicative approx. for cuts above $n^{1.5}$. It is an interesting open question of whether one can do better.
>
> > Would the techniques work for other notions of expansion?
>
> We did not consider other notions of expansion since our final goal was towards cut approximation, but this is an interesting direction for future work.

---

### Official Review · Reviewer_pbng · 2025-03-16

**Overall Recommendation:** 3

**Summary:**

This paper studies the problem of graph cut sparsification under the constraint of differential privacy (DP). They cross the known $n^{1.5}$ additive error mark to provide a DP algorithm that has an additive error of $\tilde{O}(n^{1.25+o(1)})$ and a small multiplicative error. Their key underlying subroutine is one that *privately* outputs an expander decomposition of the input graph $G$, which has not been done prior to this work. Their main algorithm has three steps, where they first run the private expander decomposition algorithm, followed by running the algorithm by Elias et al. 2020 on the inter-component edges, and finally running the algorithm by Upadhyay et al. 2021 on the individual components. There is one last post-processing step to sparsify the graph using non-private prior work (which is fine in this case as we're just post-processing). Their DP expander decomposition algorithm is of independent interest. This paper has purely theoretical results, and they show other theoretical applications of their work.

**Claims And Evidence:**

I'm confused by the proof of Theorem 3.2 It just says that it's in Algorithm 1. That's not really a proof. Or are you just trying to say that you will show this constructively by proving the correctness of Algorithm 1?

**Essential References Not Discussed:**

Not that I'm aware of.

**Experimental Designs Or Analyses:**

N/A, since the results are purely theoretical.

**Methods And Evaluation Criteria:**

The metrics make sense -- the problem is very well-defined, so there is no question of checking the evaluation criteria. Theoretical comparison with prior work is there, which is pretty much what we're looking for.

**Other Comments Or Suggestions:**

1. For Theorem 4.1, maybe explain the meanings of $d_{exp}$ and $d_{size}$. I don't know what they're supposed to mean beforehand, and what they're supposed to imply.
2. More explanation about the theorems and what their roles are would make sense. For example, some details about what Theorem 4.1 and 4.2 are saying, along with what Lemmata 4.3 and 4.4 are implying would be really helpful. It just feels like I'm reading technical statements without context/intuition on why they're going to be useful.
3. For Section 3.1, two partitions will get affected if you're doing replacement definition. Under add/remove, only one partition will be affected. Please, be clear about your model of privacy.

**Other Strengths And Weaknesses:**

Strengths:
1. The $\tilde{O}(n^{1.25+o(1)})$ additive bound is great and is significant. It's a big improvement over the prior $O(n^{1.5})$ bounds.
2. I like the DP expander decomposition algorithm as a subroutine, and I'm curious to see what other applications it may have. It seems like a good contribution -- to be fair, it might be the main contribution, given that the other two steps of the main algorithm are simply applying the prior work as black-box.
3. The theoretical applications are also interesting. Would like to see more though to put the results into perspective a bit more.

Weakness:
1. I'm mostly concerned about the writing a bit. Whilst the high-level flow is clear enough, the low-level details are sometimes hard to follow. I will write a little bit below.

**Questions For Authors:**

Have you thought about node DP for this problem? What could the potential challenges be and how would the bounds be affected? It's just a speculative question, rather than something for the purpose of evaluation.

Also, have you thought about proving lower bounds in this multiplicative error setting? I'm mostly speculating about the tightness of your bounds (mostly out of curiosity).

**Relation To Broader Scientific Literature:**

This is a well-defined fundamental problem that didn't have better results. The results in this work are significantly better, but that said, it keeps the question open to get even better bounds on the additive error. The DP expander decomposition algorithm is of independent interest with potential applications in other places.

**Theoretical Claims:**

I read the proofs quickly in the main body, but briefly glanced at the proofs in the appendix. What I looked at seemed fine.

---

> ### Author Rebuttal · Authors · 2025-04-01
>
> Thank you for your valuable feedback! We will adjust the final draft according to your suggestions, elaborating where there were points of confusion.
>
> > I'm confused by the proof of Theorem 3.2 It just says that it's in Algorithm 1. That's not really a proof. Or are you just trying to say that you will show this constructively by proving the correctness of Algorithm 1?
>
> We apologize for the confusion. Please note that the proof environment for Theorem 3.2 ends on Line 310 and the proof is contained in Subsections 3.1 - 3.3. Each subsection handles the privacy, approximation, and success probability parts of the proof separately. However, we admit it is misleading to use numbered subsections to organize a proof. We will change the style.
>
> > For Theorem 4.1, maybe explain the meanings of d_exp and d_size. I don't know what they're supposed to mean beforehand, and what they're supposed to imply.
>
> We used the same notation as in the prior work of Li & Saranurak. The intuition is that the theorem can find a cut such that any possible cut with substantially smaller sparsity (as measured by d_exp; note exp stands for expansion), must also be smaller in size in terms of the number of vertices (as measured by d_size). Note that later in Theorem 4.2 we obtain a private version of Theorem 4.1. We will update the text to include more intuition.
>
> > More explanation about the theorems and what their roles are would make sense. For example, some details about what Theorem 4.1 and 4.2 are saying, along with what Lemmata 4.3 and 4.4 are implying would be really helpful. It just feels like I'm reading technical statements without context/intuition on why they're going to be useful.
>
> Thanks for the suggestion. We will include a more intuitive summary of Section 4 before the theorem statements and proofs in the paper. At a high level, in Theorem 4.1 and Lemma 4.3, we introduce the existence of an algorithm for non-private expander decomposition from prior work. By expander decomposition, we mean a partition of the vertices such that every part has high connectivity in the sense that every cut is non-sparse (has a high fraction of edges to vertices). We use Lemma 4.4 from prior work, which upper-bounds the recursive depth of this algorithm. Theorem 4.2 introduces our contribution, which replaces the core algorithm from Theorem 4.1 with a private analogue.
>
> > For Section 3.1, two partitions will get affected if you're doing replacement definition. Under add/remove, only one partition will be affected. Please, be clear about your model of privacy.
>
> This is a good question. We use the standard model of edge-neighboring privacy, which is the same as the add/remove model over the database of edges. Importantly, two partitions are still affected in the add/remove model because a single edge touches two vertices.
>
> > Have you thought about node DP for this problem? What could the potential challenges be and how would the bounds be affected? It's just a speculative question, rather than something for the purpose of evaluation.
>
> This is a nice open question raised by Elias et al. This setting is much harder and there is nothing non-trivial known for preserving cuts using node DP to the best of our knowledge.
> > Also, have you thought about proving lower bounds in this multiplicative error setting? I'm mostly speculating about the tightness of your bounds (mostly out of curiosity).
>
>
> We are optimistic that near-linear additive error, when allowing constant multiplicative error, is possible to achieve in polynomial time. It is already known how to achieve this in exponential running time; see Eliás, Kapralov, Kulkarni, Lee SODA’20. We believe this is an interesting challenge for the field. We also note that $\Omega(n)$ error is necessary even when allowed a multiplicative approximation, see Dalirrooyfard, Mitrović, Yuriy Nevmyvaka, NeurIPS’23.

---

### Decision · Program_Chairs · 2025-05-01

**Decision:**

Accept (poster)

**Comment:**

The paper develops a private algorithm for expander graph decomposition in order to give a better private cut sparsifiers. The reviewers agree that the result is interesting and a clear improvement over prior work. Moreover, this seems to be the first application of expander decompositions to private algorithms, and it may influence future work. The paper would make a nice contribution to ICML.